# Evaluation of the Nutritional Quality of Processed Foods in Honduras: Comparison of Three Nutrient Profiles

**DOI:** 10.3390/ijerph17197060

**Published:** 2020-09-27

**Authors:** Adriana Hernandez Santana, Sharyl Waleska Bodden Andrade, Dina Rojas Aleman, Jean Pierre Enríquez, Adriana Beatriz Di Iorio

**Affiliations:** 1Human Nutrition Laboratory, Department of Food Science and Technology, Zamorano University, San Antonio de Oriente, Francisco Morazan, Tegucigalpa 11101, Honduras; adi@zamorano.edu; 2Faculty of Health Sciences, Central American Technological University, Tegucigalpa 3539, Honduras; sharyl.bodden@unitec.edu; 3Zamorano University, San Antonio de Oriente, Francisco Morazan, Tegucigalpa 11101, Honduras; dinarojas@hotmail.com; 4Masters Program in Sustainable Tropical Agriculture, Graduate Department, Zamorano University, Tegucigalpa 11101, Honduras; jean.enriquez.2018@alumni.zamorano.edu

**Keywords:** Chile, fast food, nutrients, obesity, sodium

## Abstract

Obesity is considered a global pandemic. Different countries have worked to implement front-of-package (FOP) labeling systems that define thresholds for critical nutrients (CN) as part of their public health policies. The objective of this study is to identify the proportion of Processed (PF) and Ultra-Processed (UPF) Foods marketed in Honduras, which meet or fall short of the criteria of three Nutrient Profile Models (NPM): PAHO (2016), Chile (2017) and the Central American Technical Regulation Proposal-Nutritional Warning Front Labeling (CATRP-NWFL 2017). This study is descriptive; 1009 products from 206 brands were collected nationwide. Descriptive statistics were performed. The mean CN compliance with the three models was 49.3% for sodium, 30.6% for sugars, 63.1% and 96% for saturated and trans fats. The PAHO and Chilean (NPM) similarly concentrated on the lower compliance with the established criteria, unlike the CATRP, which has less stringent criteria. This is the first assessment of CN content in PF and UPF in Honduras under three different NPMs. We highlight the importance of defining or adopting criteria for the implementation of NWFL as information for the consumer and thereby contribute to reducing the risks of obesity and related diseases.

## 1. Introduction

Globally, the number of people with obesity has tripled in the past 40 years [1,2]. The latest report from the World Health Organization (WHO) considers overweight and obesity as a global pandemic. In 2016, more than 1.9 billion adult people from age 18 and over were overweight; of these individuals, more than 650 million were obese. The prevalence of overweight and obesity among children and adolescents aged 5–19 has risen dramatically from 4% in 1975 to 18% in 2016 [2]. Moreover, the highest proportion of overweight and obesity corresponds to the countries affected in the Pacific and the Caribbean, countries of the Middle East and Central America. At the Latin American level, the highest proportion of obesity among boys is in to Chile (11.9%) and Mexico (10.5%), whereas the highest obesity rates for girls are observed in Uruguay (18.1%) and Costa Rica (12.4%) [3].

One of the determining factors for the global increase in obesity is the frequency of consumption of processed (PF) and ultra-processed (UPF) foods, causing alteration in people’s diet. These have been defined by the Pan American Health Organization (PAHO) as “industrial products that are characterized by having high caloric density together with high levels of sugar, saturated fats, sodium, and deficient levels of vitamins and minerals”. While PFs are mostly made up of two or three ingredients, UPFs can have five to twenty, or more, including substances that have been extracted from foods of unusual culinary use; substances made from food components; and additives used for modifications in color, flavor, taste and texture of the final product [4]. Some examples include sweet or salty packaged snacks, chocolates, candies, cookies, breakfast cereals, jams, jellies, carbonated and energy drinks, milk-based sugary drinks, and more [4,5].

Scientific evidence supports the strong association between UPF consumption and the risk of developing Non-Communicable Diseases (NCDs), especially obesity [5,6,7,8,9]. The prevalence of these conditions increases directly as the consumption of UP and UPF surges [10]. Some studies have shown that, in high-income countries, the consumption of processed foods and beverages provides for more than two-thirds of dietary energy [11]. Similarly, in low and middle-income countries, the consumption of these foods is increasing [12,13].

During the 2000–2013 period, a significant increase in the volume of sales of ultra-processed foods and beverages could be observed, considering Latin America as the potential market for this range of products, except for Argentina and Venezuela. Modern grocery stores, especially supermarkets and hypermarkets, tend to dominate the distribution channels for this type of food [5].

Research on the influence of eating habits and nutrition amongst the Honduran population is scarce for both children and adults. A study carried out in the municipality of San Antonio de Oriente (Honduras) identified that the consumption of energy-dense foods begins from an early age, observing deficiencies in the consumption of fruits and vegetables, followed by an excess of sodium intake throughout the subjects’ life-course, concluding that they suffer from a high nutritional risk and involvement of NCDs [14]. This situation highlights the need to implement policies aimed at informing consumers regarding the high content of calories, sugars, sodium, and saturated fats in foods and beverages. Reducing the intake of these products may prevent related NCDs due to an inadequate diet.

In order to guide countries in the prevention of obesity and other related diseases, PAHO published its Nutrient Profile Model (PAHO-NPM) [5] in 2016, which it defines as “the science of classifying foods according to its nutritional composition for reasons related to disease prevention and health promotion” [15].

The Nutrient Profile Model (NPM) can be used by national authorities to formulate policies such as restrictions on advertising to children, the inclusion of front labeling or health and nutritional claims [16]. Different countries have worked on the implementation of front-of-package (FOP) labeling systems, establishing criteria or thresholds on the content of Critical Nutrients (CN) to include specific warnings on the packaging of products that exceed the established thresholds [17].

A recent study carried out in Honduras (2018) on the content of CN in 520 PF and UPF available in the Honduran market reported that 75% of the products, analyzed under the PAHO-NPM, presented excess sugars, (37%) sodium, (33%) total fat, and (30%) contained sweeteners [18] per serving.

Chile in 2016 was the first country to implement a mandatory warning label system (Chilean-NPM) for foods high in caloric density, added sugar, sodium, and saturated fat with various stages of implementation that were finalized in 2019 [19,20,21]. Mexico, as part of a government strategy to prevent obesity and NCDs, has implemented the proposal for mandatory front labeling, with a special focus on providing easy-to-understand information on ingredients with a negative impact on health (added sugars, sodium, total fats, saturated fat) and caloric density, which facilitate the most convenient choice being made according to the health of each person [22].

Recently, the Institute of Nutrition of Central America and Panama (INCAP) has proposed the Central American Technical Regulation Proposal (CATRP) of “Food and Drinks Front Labeling of Nutritional Warning (FLNW): Requirements for its application” [23], based on the PAHO-NPM (2016), with minor variations. Moreover, this is the first study where the INCAP proposal is taken into consideration.

Honduras does not have an NPM or the obligation to include the nutritional label on food packaging, and since the current Central American Technical Regulation Proposal is voluntary [23], the CATRP-FLNW requires more evidence for its implementation. To date, the proportion of PF and UPF marketed in Honduras where CN content meets or disregards the criteria established by the PAHO-NPM or any other model, such as the Chilean-NPM and the CATRP-FLNW, is unknown, since Honduras imports most of these types of products. Mexico carried out this exercise with seven different NPMs [24] and observed differences between the food categories according to each NPM, with the PAHO-NPM being the most rigorous.

The objective of this study is to identify the proportion of PF and UPFs with excessive amounts of CN by using three nutrient profile models: PAHO-NPM, Chilean-NPM and CATRP-FLNW, and to compare their nutritional content among the products that meet, or not, the criteria established for each NPM. The information from this analysis will provide further evidence to drive the implementation of the CATRP-FLNW for the benefit of the Honduran population.

## 2. Materials and Methods

The descriptive study was carried out in the human nutrition laboratory of Zamorano University, located in the department of Francisco Morazán, Honduras. The products were collected from the main supermarket chains located in the two most important cities in Honduras (Tegucigalpa and San Pedro Sula).

### 2.1. Data Collection

The data collection of the nutritional content of PF and UPF was carried out between July and September 2019 by a team of qualified professionals. The 2018 product database was updated, in which information was added on new products by purchasing and taking photographs of the labeling and packaging. Subsequently, a stratification was performed according to their nutrient profile in an Excel 2013 datasheet.

The PAHO-NPM and the NOVA classification were used as a reference to define the products in minimally processed, processed, and ultra-processed foods [5,25]. The categories selected usually included PFs and UPFs, which has been one of the largest discussions on nutrition policy [25].

For the selection of the products, whether they included information of the company, brand, country of origin, nutritional information, container size, and list of ingredients was taken into consideration. The products were classified into the following categories: non-dairy beverages, dairy products, salty snacks, sweet snacks, cereal products, bread and bakery, and various products. Food supplements and culinary ingredients, such as cooking oils, butter, salt, honey, sugar, sweeteners, and freshly prepared dishes, were excluded from the analysis [5,25]. Only PF and UPFs were considered for their respective analysis of compliance with the CN thresholds of each model used in the present study.

It was verified that the nutrient content of each product in the database was complete and correctly entered; any incomplete information was cross-verified with the photographic database. A random review of 5% of the products was carried out to ascertain the quality of the data, and no irregularities were found. When the nutritional label did not report the amount of free sugars, the method proposed by PAHO was used to calculate the free sugars based on the total declared sugars in the packages [5].

Duplicate or missing data was excluded from the analysis (*n* = 119), as well as those in which data errors were identified, or if the products were replicated, such as the same product in different presentations. The detailed breakdown of the excluded products is described in Figure 1. In this way, information on 1009 products (PF and UPF) corresponding to 206 brands present in the Honduran market was analyzed.

### 2.2. Nutrient Profiling Systems

Three NPMs used for Latin American countries were used to compare the nutrient content of PF and UPF:(1)Nutrient Profile Model of the Pan American Health Organization/PAHO (PAHO-NPM).(2)Chilean Nutrient Profile Model (Chilean-NPM).(3)The Central American Technical Regulation Proposal for “Frontal Labeling of Nutritional Warnings” (CATRP-FLNW) by INCAP (Institute of Nutrition of Central America and Panama).

The PAHO-NPM served as the basis for the creation of the Chilean-NPM and CATRP-FLNW, used in the countries’ food policy [26]. Each product was individually classified according to the criteria of each NPM, verifying the presence of sodium, sugar, saturated fats, and trans fats in particular (Table 1).

### 2.3. Statistical Anlysis

As part of the analytical statistics, the mean and standard deviation (SD) were estimated for each of the food groups present in the three NPMs. Confidence intervals with 95% reliability were obtained using the SAS version 9.4 program. The nutritional content of food and beverage categories was examined, and the percentage of products that did, or did not, meet the criteria for each nutrient limit was estimated. The Chilean-NPM has established thresholds to determine compliance or non-compliance with a permissible limit. For the PAHO-NPM and CATRP-FLNW, the analysis was performed based on the maximum allowable cut-offs for each product, according to the criteria mentioned in Table 1.

## 3. Results

Foods were cataloged into seven categories and classified as: non-dairy beverages (*n* = 208), dairy (*n* = 71), salty snacks (*n* = 99), sweet snacks (*n* = 235), cereal products (*n* = 182), bread and bakery (*n* = 48), and various (*n* = 166). Their descriptions are presented in Table 2.

The highest percentage of analyzed products came from the American continent, and the lowest proportions from Asia and Europe, as can be observed in category in Figure 2. By category, the majority of products from Asia are non-dairy beverages, and those from Europe are sweet snacks.

It is possible to identify PF and UPF where CN content complies with the thresholds of each of the models mentioned in Table 3, and estimate those found with excessive CN content. For example, 44.2% of the total products comply with the PAHO-NPM, in respects to sodium.

By NPM, of all the products analyzed under the PAHO-NPM, 44.2% met the threshold allowed for sodium, 25.0% for free sugars, 60.5% for saturated fats, and 94.9% of the total products analyzed in this study were within the established for trans-fats. This means that the lowest compliance with the established criteria corresponds to the content of free sugars (75.0% of the products do not meet the criteria). Bread and bakery (0%), non-dairy beverages (2.8%), dairy products (11.2%) and cereal products (12%) presented the lowest compliance percentages, which is why they constitute the PF and UPF categories with excessive contents of free sugars. Salty snacks were the products with the highest compliance percentage (86.8%) according to free sugar content, followed by the various product category (57.8%).

After the lower compliance in the content of free sugars applying the PAHO-NPM, the proportion of PF and UPF corresponded to an excessive content of sodium (55.8% of non-compliance). Most of the various products (90.3%) met the sodium criteria; bakery products, sweet snacks, dairy and non-dairy beverages were the categories with the least compliance; that is, they evinced excessive sodium contents.

Non-dairy beverages and cereal-based products met the criteria for saturated fat by more than 85% (94.2% and 89.5%, respectively). The bread and bakery category presented the lowest percentage of compliance for saturated fat (2.0%), in comparison to the other categories of products analyzed that obtained a higher percentage of compliance.

In the framework of the Chilean-NPM (Table 3), 31.4% of the products met the criteria for sodium, 60.3% for total sugars, and 32.6% of the products were within the established range for saturated fats. The categories of salty snacks, various products, bakery, and cereal products were the categories with the lowest compliance in respect to the limits established regarding sodium (25.2%, 27.7%, 45.8%, and 50%, respectively) with the Chilean model.

Additionally, low levels of compliance with total sugar criteria were also observed in the categories of sweet snacks (12.7%), cereal products (14.2%), dairy beverages (16.9%), and non-dairy beverages (22.5%). The majority (99.5%) of non-dairy beverages, dairy beverages (98.5%) and cereal products (85.7%) were within the established range for saturated fats. However, low percentages of compliance were identified in the categories of salty snacks, sweet snacks, bread and bakery (21.2%, 28.1%, and 47.9%, respectively).

With the CATRP-FLNW model (Table 3), for the total of analyzed products, only 43.3% met the sodium criteria, 34.2% for total sugars, 63.6% for saturated fat, and 97.2% for trans-fat.

The proportion of products that met the CATRP-FLNW criteria for total sugars represents a minimum percentage in the categories of non-dairy beverages (6.7%), dairy beverages (11.2%), bread and bakery (14.5%), sweet snacks (21.6%) and cereal products (25.8%). Most products met the criteria for saturated fat in the categories of non-dairy beverages and cereal products (98.5% and 89.5%, respectively), unlike salty snacks (32.3%), dairy beverages (35.2%), and sweet snacks (38.7%).

Dairy beverages, sweet snacks, and non-dairy beverages had low compliance in regards to the established sodium thresholds (18.3%, 20.8%, and 22.5%, respectively) in this model. The products with scant compliance with the criteria of total sugars are non-dairy beverages, dairy beverages, and bread and bakery (6.7%, 11.2%, and 14.5%, respectively).

Regarding the values of CN content in food products that were obtained by applying the reference models, the average content of each CN was lower in compliant products than in non-compliant products, as shown in Table 4.

The differences between the three models analyzed are statistically significant for total sugars (*p* < 0.05), sodium (*p* < 0.05), and free sugars (*p* < 0.05). These were the CN that showed the greatest difference in the mean contents of the food categories, followed by saturated fat (*p* < 0.05). Whereas, for trans fats in the PAHO-NPM, the nutritional content was approximately similar, without significant differences.

In Table 4, using the Chilean NPM, the products that met the threshold of the three CN presented a lower average compared to those that did not meet the threshold. However, products that complied with PAHO-NPM and CATRP-FLNW models, particularly sodium (*p* < 0.05), presented a higher mean compared to those that did not comply. Therefore, the Chilean model was stricter because it presented less compliance when compared to the other models.

Table 5 shows the percentages of food products that generally complied with all the criteria according to the applied NPM. In general, the PAHO-NPM (8.0%) was as strict as the Chilean-NPM (8.8%), whereas CATRP-FLNW was the least strict (11.6%) of the three models, specifically in the cereal categories, bread and bakery, and various products.

## 4. Discussion

We observed clear differences in the average nutritional content for each CN, especially for sodium, saturated fat, free and total sugars. The WHO recommends limited consumption of CN [27,28,29]. In Honduras, there have been no previous studies on the proportion of products available under various nutrient-profiling systems, as it has been done by Colombia [30] and Mexico [24], who reported similar results to this study.

The PAHO-NPM is the reference model since it is the one that proposes the strictest criteria in the CN content; therefore, it was the model capable of identifying those products that met these criteria, being the one that identifies the least compliance ratio in all CNs (8.0%).

The greatest disparity in terms of the proportion of products with excess in CN content, according to the criteria of the three models, was identified in sodium, followed by sugars, in which the three models had different criteria. The PAHO-NPM refers to free sugars (if the calories from free sugars were equal to or greater than 10% of the total calories), Chilean-NPM to total sugars (if per 100 mL of liquids it was greater than 5 mg; or 10 g per 100 g in solid products) and the CATRP-FLNW to total sugars (if it was greater than or equal to 20% of the total energy from sugars totals). A similar situation can be observed with saturated fats, where the PAHO-NPM was the strictest. Regarding trans-fat, the PAHO-NPM considers excess trans-fat if the calories from trans-fat were equal to or greater than 1% of the total calories. Meanwhile, the CATRP-FLNW did not accept the presence of any amount of trans-fat in its formulation.

In the analysis of Table 5, it can be seen from this sample that Honduran food does not meet the recommendations for Critical Nutrient intake, with the bakery category being the most critical in terms of its CN content. It is noteworthy that this category refers to broad artisanal production, being a micro-enterprise category, which requires regulation and training.

According to Table 6, the average of the three models was 49.3% for PF and UPF, which met the criteria for sodium content, and the remaining 50.7% had excessive contents of this critical nutrient. Particularly, a higher non-compliance percentage was observed among the sweet snacks category; however, the highest non-compliance was found in dairy beverages, according to the CATRP-FLNW (18.3% of compliance). In general, the highest compliance in all the categories was observed in the Chilean-NPM (60.3%), and the PAHO-NPM and CATRP-FLNW were similar to each other.

For sugars, Table 6 shows that, the average of the three models showed 30.6% for the PF and UPF met the sugar content criteria, whether they were free (PAHO-NPM) or total (Chilean-NPM and the CATRP-FLNW). Therefore, the remaining 69.4% bear an excessive content of this critical nutrient, especially higher percentages in non-dairy beverage category, with only 2.8% compliance according to the PAHO-NPM. In general, the highest compliance for the total of the categories was affected by the CATRP-FLNW (34.2%), followed by the Chilean-NPM (32.6%).

For saturated fats, Table 6 shows that the average of the three models, 63.1% of PF and UPF, met the criteria for saturated fat content, corresponding to the highest non-compliance ratio in salty snacks, followed by sweet snacks and the bread and bakery category, with the lowest compliance value (2.0%), followed by non-dairy drinks for free sugars (2.8%), according to the PAHO-NPM. In general, the highest compliance in all food categories was observed in the Chilean NPM (65.1%), which was close to the proportion estimated with the CATRP-FLNW (63.6%). Salty snacks was the category with the lowest compliance regarding saturated fat content (28.9%).

The category labeled as sweet snacks showed the lowest compliance for the three NPMs with respect to sodium, sugars, and saturated fat content. In the same way, non-dairy drinks, followed by dairy drinks, cereal products, and sweet snacks, were the product categories with the lowest compliances, in regards to sugar content in the three models. Bakery products were the foods with the lowest compliance with the trans-fat content criterion, according to the PAHO-NPM, which suggests that the offer of this type of product to Honduran food does not meet the recommendations for this CN.

Observing the analysis of the three models in Table 6, there is a clear trend regarding compliance in the bakery and sweet snack categories with respect to sodium limits, with an average of less than 43%. The above is of great interest because most of the breakfast and snack foods provided to school-age children fall within these categories. Therefore, this generates concern regarding the development of bad eating habits, increasing the rates of childhood overweight and obesity, along with the future adults with a higher risk of NCDs. This observation does not only apply to schoolchildren in low-income areas, but it is also observed in middle and upper-class areas.

A cross-sectional study carried out in Canada [31] analyzed foods and beverages with low nutritional quality under various NPMs and showed that, in general, the PAHO-NPM presented the strictest criteria, which was only met in 9.8% of foods. Nevertheless, they were classified as the most eligible for marketing to children. These results are consistent with the present study, especially in the categories of bakery products, snacks, beverages, and dairy products, which present low compliances. In addition, it is necessary to define strategies aimed at improving the supply of better nutritional quality food in national markets with the proposal to reduce the supply of PFs and UPFs, as well as promoting the reform of products to improve their nutritional quality.

Studies carried out in Mexico [32] and France [33] reported that the nutritional content of cereal products had great variability among all CNs with different NPM, particularly for sugars, saturated fats, and sodium, which did not concur with the results of this study. Regardless of the NPM used, this category showed similar results at the different CNs.

Moreover, another global study [34] on cereal products revealed a high content of CN such as sugar and sodium. This finding is consistent with the results of the present study, where low compliance prevails concerning sodium and sugars, and cereals were found to provide low nutritional quality in their formulation.

In Honduras, the analysis of a national sample of 144 commercial processed foods, determined to be functional, showed that sweetened breakfast cereals were the most caloric (180 kcal) per serving.

More than 70% of functional processed foods exceeded the sugar recommendation with the PAHO-NPM, mainly including sweetened breakfast cereals (100%) and sweetened juices (93%). Regarding sodium, 49% of the products exceeded the recommendation, certain baked products (100%) as well as cheeses (86%), just like 100% of the products exceeded the total fat recommendation. In that study, more than 60% of the products were selected simultaneously and exceeded between 2 and 3 PAHO-NPM criteria; 5% exceeded all the criteria and 4% met all the PAHO-NPM criteria [35]. In the present study, 8.0% met all the criteria for this model, without being classified as functional, which points to the urgent need to implement the respective regulations for the timely identification of products with excess in one or more CN.

A fundamental difference between the use of the PAHO-NPM and the CATRP-FLNW concerning the Chilean-NPM is the use of an approach based on nutrient density, concerning the volume of nutrients to classify foods. The former is based on energy density (nutrients per calorie), unlike the Chilean-NPM that calculates the content of nutrients per 100 g or 100 mL of the total products (nutrients in relation to volume) [21].

With the application of the PAHO-NPM, low-calorie density products could be regulated due to a high ratio of nutrients compared to calories. For example, for sodium, the PAHO-NPM is based on a 1:1 ratio (1 mg sodium/1 kcal), unlike the Chilean-NPM, in which the criteria being analyzed is based on sodium density, representing variations in respect to the type of product, that is, whether it belongs to the food or beverage category. Therefore, under the PAHO-NPM, product categories that are low in calories but high in sodium could exceed the thresholds for this critical nutrient [5]. Hence, sodium was the CN with the greatest variability in the three models, identifying that the Chilean-NPM is more permissive.

This study provides comparative evidence of what could happen with the implementation of the proposed CATRP-FLNW for the Central American region and review of the advisability of proposing stages of implementation to adjust these criteria. This situation is essential for the benefit of the population and to control the current NCDs epidemic, in addition to implementing regulations for zero use of trans fats in products to prevent cardiovascular disease [36]. From the models applied to this study, it can be seen that CATRP-FLNW would require strengthening its regulation towards sugars, since more permissiveness is observed with this specific CN, probably gradually and with the industry’s acceptance. Its application must be mandatory and for the benefit of the population.

A nutrient profile with weak criteria is on par with the classification of a higher percentage of processed foods as “regulated”, therefore it may be necessary to promote the intake of CN in lower amounts in the diet [19], with the respective implications. However, a nutrient profile with very strict criteria would not represent an attractive option for the food industry, and consequently, its efficiency in food promotion and regulation would be limited [37], unless there was a systematic reduction of CN, as seen in the example of Chile [20].

The products that present mostly excess contents of various critical nutrients should be regulated in their offer and their advertising, as some countries have already done, as well as the implementation of frontal labeling, such as in Chile [20], Ecuador [38] and France [39], and thus be able to include dietary guidelines to promote a healthy and balanced diet, allowing the inclusion of prefabricated foods in a healthy and responsible way [40]. This should include basic studies regarding the consumption of nutrients in the population to identify the existence of a positive impact on the practice of food selection based on its nutritional quality. The results obtained in this study can be useful for the elaboration of standardized educational material aimed at the school population to improve and modify the needs and eating styles of the Honduran population.

Equally important is the updating of the dietary guide, which would be based on the results of this study and the CATRP-FLNW, and if necessary, carry out more studies to establish the usefulness of the criteria for each model of interest [40,41], taking into account the target population, vulnerable population and knowledge of nutritional topics (diet and food availability), focused on improving public health [25], identifying agreements with the food industry, in light of the urgent need to promote reformulation of products [42] and regulation of advertising [43,44].

It is known that FLNW plays a priority role in the area of health, considering that not all FLNW contain abundant and detailed information on the nutritional value of the product, or do not convey data that facilitates the consumer’s understanding of the percentage of nutrients, and the value in the consumption of each one. Thus, there is a need for research on how FLNWs work in the real world. Does the population understand the values reflected in the FLNW? Does the FLNW create a real impact when making the purchase of food by the population [45]?

The limitations of this study include the inability to count the regulatory agents in Honduras regarding the limits of the critical nutrient content and control in the production of PF and UPF. There was also no front labeling implemented in national production; this highlights the importance of supporting the implementation of the CATRP-FLNW.

The strengths of this study rely on the information collected directly from Honduran supermarkets at various levels and locations. The sample was made up mostly of PFs and UPFs of Latin American origin that could be evaluated by the three nutrient profiles, although there are others worldwide, and those with relevance at the Latin American level were included. Analysis preference was given to the related critical nutrients of greatest interest, such as the content of sodium, sugars, and saturated fats.

The study used the PAHO-NPM as a reference, which has been previously validated using specific methods that show greater validity in the identification of foods with the presence of excess critical nutrients, which the CATRP-FLNW uses as a basis, as well as the experience from countries like Chile.

This study promotes the continuation of CN content analysis to support the implementation of the CATRP-FLNW for Honduras and the region, incorporating the strengths of the applied models.

## 5. Conclusions

Updated evidence was generated on the content of CN in PF and UPF in Honduras under three different nutrient profiles. The lowest compliance was found in sugar content, followed by sodium and saturated fats, and the best compliance was achieved by trans fats.

The PAHO model is the reference model since the products evince the lowest compliance when applying its criteria. The results confirm the important need to implement mandatory front labeling on the container, due to the low proportion of products that meet the criteria of the three NPMs. It is an important and useful tool to define and monitor actions and policies aimed at improving the quality of products on the market, raising awareness of the consumption habits of the population, and reducing the risks of obesity and NCDs.

The intervention of public health institutions is essential to focus special attention to promoting FLNW. With the population’s consumption of PF and UPF on the rise, the need for raising awareness of CNs is growing exponentially. This analysis highlights the importance of reform in products containing critical nutrients for the health of the population.

## Figures and Tables

**Figure 1 ijerph-17-07060-f001:**
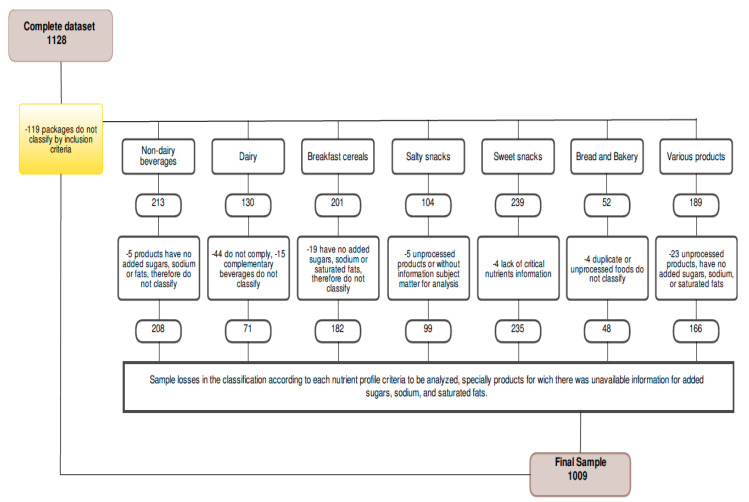
Description of PF and UPF dropouts: flow diagram. Products without critical nutrients information, or duplicated, were excluded from the final sample. Source: Made by the authors.

**Figure 2 ijerph-17-07060-f002:**
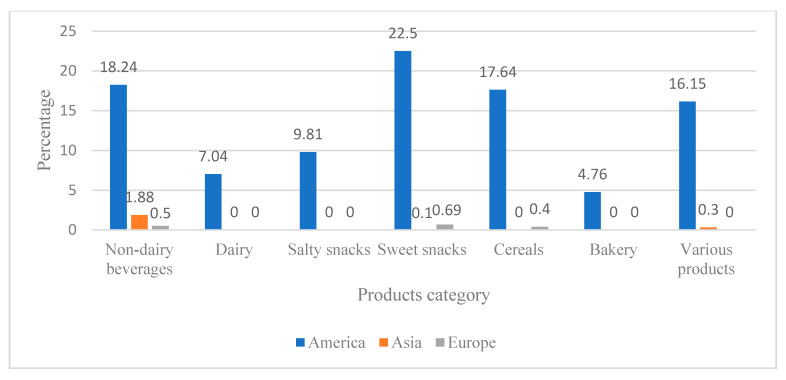
Percentage distribution of products (PF and UPF) by category according to origin by continent. The American continent had the majority of products, followed by Europe and Asia. Source: Made by the authors.

**Table 1 ijerph-17-07060-t001:** Criteria for critical nutrient profiles (sodium, sugar, saturated, and trans fats) according to the PAHO-NPM, Chilean-NPM, and CATRP-FLNW model.

Nutrient Profile	Reference Quantity	Sodium Reference Quantity	Total Sugars	Free Sugar	Saturated Free Fats	Trans Fat
Pan American Health Organization (PAHO, 2015) [5].	Per kcal of energy	Excess sodium if the ratio of sodium (mg) to calories is equal to or greater than 1:1	N/A	Excess free sugars if calories from free sugars are equal to or greater than 10% of total calories	Excess saturated fat if calories from total fat are equal to or greater than 10% of total calories	Trans fat excess if calories from trans fat are equal to or greater than 1% of total calories
Chilean NPM [19,20].	Per portion of the product; expressed by 100 g or 100 mL	Excess sodium mg if per 100 mL of liquids it is greater than 100 mg or greater than 400 mg per 100 g in solid products	Excess sodium mg if per 100 mL of liquids it is greater than 100 mg or greater than 400 mg per 100 g in solid products	N/A	Excess saturated fat if per 100 mL of liquids it is greater than 3 mg; or 4 g per 100 g in solid products	N/A
Proposal INCAP CATRP-FLNW [23].	By energy content of the product expressed per 100 g or 100 mL or per serving	Excess sodium if the ratio of the amount of sodium mg to calories is equal to or greater than 1:1	Excess of total sugars if it is greater than or equal to 20% of the total energy from total sugars	N/A	Excess saturated fat if it is greater than or equal to 10% of the total energy from saturated fat	Presence of any amount of trans fat

Depicted above are the thresholds and nutritional objective for each of the nutrient profile analyzed. NA: Not Applicable. Source: Elaborated by the authors.

**Table 2 ijerph-17-07060-t002:** Number and proportion of products (PF and UPF) classified by food categories.

Food Category and Classification	*(n)* Sample	%
**Non-dairy beverages**	**208**	**20.6**
Juices, beverages and nectars	132	63.5
Powders to prepare sugary flavored drinks	16	7.7
Energy drinks	8	3.9
Carbonated beverages	29	13.9
Flavored teas	12	5.8
Zero calories	4	1.9
Vegetable milk (soy, almond, rice)	7	3.4
**Dairy**	**71**	**7.0**
Yogurt	40	56.3
Pudding	5	7.0
Flavored milk with added sugar	20	28.2
Coffee with milk beverages	6	8.5
**Salty snacks**	**99**	**9.8**
Packaged chips	82	82.8
Dried fruits with added salt	17	17.2
**Sweet snacks**	**235**	**23.3**
Cookies	147	62.6
Chocolates	45	19.2
Candy	43	18.3
**Cereal products with added sugar**	**182**	**18.0**
Granola	10	5.5
Oats	17	9.3
Breakfast cereals	113	62.1
Cereal bars	42	23.1
**Bread and bakery**	**48**	**4.5**
Bread with added salt	21	43.8
Sweet bread	27	56.3
**Various products**	**166**	**16.5**
Canned meat	22	13.3
Dressings and sauces	33	19.9
Flour and pasta mixes	36	21.7
Soups and creams	23	13.9
Canned seafood	11	6.6
Canned fruits and vegetables	8	4.8
Jams and jellies	15	9.0
Packed ground beans	3	1.8
Cheeses with added salt	15	9.0
Total	1009	100

The proportions of products for categories (in bold format) and subcategories are depicted below. Source: Elaborated by the authors.

**Table 3 ijerph-17-07060-t003:** Proportion of food products (PF and UPF) that meet the nutritional criteria according to PAHO-NPM, Chilean-NPM and CATRP-FLNW, by food category.

NPM	Category	Sodium		Sugars *		Saturated Fat		Trans Fat	
		*n*	%	*n*	%	*n*	%	*n*	%
	Non-dairy beverages (*n* = 208)								
PAHO		53	25.4	6	2.8	196	94.2	208	100
Chile		204	98.0	47	22.5	207	99.5	NA	NA
CATRP-FLNW		47	22.5	14	6.7	205	98.5	206	99.0
	Dairy beverages (*n* = 71)								
PAHO		18	25.3	8	11.2	30	42.2	71	100
Chile		62	87.3	12	16.9	70	98.5	NA	NA
CATRP-FLNW		13	18.3	8	11.2	25	35.2	71	100
	Salty snacks (*n* = 99)								
PAHO		62	62.6	86	86.8	33	33.3	99	100
Chile		25	25.2	84	84.8	21	21.2	NA	NA
CATRP-FLNW		61	61.6	91	91.9	32	32.3	98	99.0
	Sweet snacks (*n* = 235)								
PAHO		51	21.7	35	14.8	86	36.5	231	98.0
Chile		158	67.2	30	12.7	66	28.1	NA	NA
CATRP-FLNW		49	20.8	58	21.6	91	38.7	219	93.2
	Cereal products (*n* = 182)								
PAHO		102	56.0	22	12.0	163	89.5	182	100
Chile		91	50.0	26	14.2	156	85.7	NA	NA
CATRP-FLNW		96	52.7	47	25.8	163	89.5	181	99.5
	Bread and bakery (*n* = 48)								
PAHO		10	20.8	0	0.0	1	2.0	46	95.8
Chile		22	45.8	20	41.6	2	47.9	NA	NA
CATRP-FLNW		30	62.5	7	14.5	25	52	46	95.8
	Various products (*n* = 166)								
PAHO		150	90.3	96	57.8	101	60.8	164	98.7
Chile		46	27.7	110	66.2	114	68.6	NA	NA
CATRP-FLNW		141	84.9	120	72.2	101	60.8	160	94.8
	Total products (*n* = 1009)								
PAHO		446.0	44.2	253.0	25.0	610.0	60.5	958.0	94.9
Chile		317.0	31.4	608.0	60.3	329.0	32.6	NA	NA
CATRP-FLNW		437	43.3	345	34.2	642	63.6	981	97.2

* Free sugars for PAHO-NPM, Total sugars for Chilean-NPM and CATRP-FLNW. Percentages and proportions of products that met the nutritional criteria for each category. NA: Not applicable. Source: Elaborated by the authors.

**Table 4 ijerph-17-07060-t004:** Mean content of critical nutrients in food products that met, or did not, the nutritional criteria by nutrient profile and by food category.

Critical Nutrient for Each NPM	Non-Dairy Beverages (*n* = 208)		Dairy Beverages (*n* = 71)		Cereal Products (*n* = 182)		Bread and Bakery (*n* = 48)	
	Mean (SD)		Mean (SD)		Mean (SD)		Mean (SD)	
	C	NC	C	NC	C	NC	C	NC
PAHO								
Sodium (mg)	95.7 (184.6)	33.9 (28.2)	150.3 (86.8)	83.6 (33.0)	172.1 (53.3)	83.9 (44.2)	172.8 (77.3)	149.5 (67.5)
Free Sugars (g)	0.0 (0)	23.5 (13.0)	1.0 (1.47)	9.0 (4.2)	1.5 (1.3)	9.5 (3.3)	0 (0.0)	7.2 (5.9)
Saturated Fat (g)	0 (0.2)	0.4 (0.8)	0.6 (0.6)	2.5 (1.2)	0.3 (0.4)	2.2 (0.8)	1.5 (0.0)	0.7 (0.5)
Trans Fat (g)	0.0 (0.03)	NA	0.0 (0)	NA	0.0 (0)	NA	0 (0.0)	2.3 (1.0)
Chile								
Sodium (mg)	15.4 (13.0)	444.8 (273.2)	51.7 (17.4)	275.8 (324.6)	259.9 (115.7)	546.1 (120.7)	254.0 (102.2)	601.9 (368.2)
Total Sugar (g)	2.2 (1.5)	11.3 (7.3)	2.0 (1.97)	17.6 (33.9)	5.2 (4.0)	30.0 (9.0)	5.6 (1.3)	28.3 (8.2)
Saturated Fat (g)	0.0 (0)	10 (0)	1.2 (0.8)	29 (0.0)	0.8 (1.1)	7.15 (5.5)	1.5 (1.1)	8.0 (3.1)
CATRP-NWFL								
Sodium(mg)	107.9 (193.8)	33.40 (27.2)	159.3 (98.6)	85.5 (34.5)	173.1 (51.9)	88.9 (50.0)	188.2 (141.7)	101.1 (50.1)
Total, Sugar (g)	4.3 (4.4)	23.9 (12.3)	2.1 (2.9)	18.5 (7.5)	3.4 (2.5)	10.3 (2.8)	0.9 (0.2)	8.1 (5.8)
Saturated Fat (g)	0.0 (0.2)	1.5 (0.8)	6.8 (29.8)	2.4 (1.2)	0.3 (0.4)	2.2 (0.8)	0.6 (0.5)	3.3 (1.8)
Trans Fat (g)	0 (0.0)	0.3 (0.3)	0 (0.0)	NA	0 (0.0)	0.03 (0)	0 (0.0)	2.3 (1.0)

Mean content of non-dairy beverages, dairy beverages, cereal products, bread and bakery among compliant products (those that met the criteria according the nutrient profile studied, and the critical nutrient analyzed) and non-compliant products (which did not meet the established criteria according to the nutrient profile models used). Cells in grey indicate statistical differences (*p* < 0.05) in the mean content between compliance (C) and non-compliance (NC). NA: Not applicable. Source: Elaborated by the authors.

**Table 5 ijerph-17-07060-t005:** Percentage distribution of processed foods by category that met all the criteria in each NPM.

Model	Overall (*n* = 1009)	Non-Dairy Beverages (*n* = 208)	Dairy Beverages (*n* = 71)	Salty Snacks (*n* = 99)	Sweet Snacks (*n* = 235)	Cereal Products (*n* = 182)	Bread and Bakery (*n* = 48)	Various Products (*n* = 166)
**PAHO**	8.0%	1.9%	1.4%	14.1%	5.1%	5.5%	0%	24.1%
**Chile**	8.8%	21.2%	15.5%	2.0%	0.9%	5.5%	4.2%	10.8%
**CATRP**	11.6%	3.4%	1.4%	15.2%	4.3%	13.2%	14.8%	31.9%

Products that met the thresholds established by the nutrient profile models analyzed. Source: Elaborated by the authors.

**Table 6 ijerph-17-07060-t006:** Summary of the proportion of food products by category that meet nutritional criteria, according to the PAHO-NPM, Chilean-NPM and the CATRP-FLNW.

Critical Nutrient	Category	Non-Dairy Beverages (*n* = 208)	Dairy Beverages (*n* = 71)	Salty Snacks (*n* = 99)	Sweet Snacks (*n* = 235)	Cereal Products (*n* = 182)	Bread and Bakery (*n* = 48)	Various Products (*n* = 166)	Total (1009)
	NPM	% (95% CI)	% (95% CI)	% (95% CI)	% (95% CI)	% (95% CI)	% (95% CI)	% (95% CI)	% (95% CI)
Sodium	PAHO	25.4 (51.1, 164.9)	25.3 (118.0, 196.9)	62.6 (179.6, 247.0)	21.7 (70.7, 127.1)	56 (161.6, 182.6)	20.8 (114.7, 180.3)	90.3 (380.9, 561.1)	44.2 (226.4, 294.4)
	Chilean	90 (13.6, 17.2)	87.3 (47.3, 56.2)	25 (160.4, 280.7)	67.2 (102.2, 129.5)	50 (235.9, 284.1)	45.8 (209.8, 299.4)	27.7 (135.3, 221.4)	60.3 (101.6, 121.3)
	CATRP-FLNW	22.5 (51.1, 164.9)	18.3 (99.7, 219.0)	61.6 (183.1, 250.2)	20.8 (65.8, 124.8)	52.7 (162.7, 183.7)	62.5 (135.3, 241.2)	84.9 (404.6, 592.6)	43.3 (233.9, 303.4)
Mean		48.6	43.6	49.8	36.6	52.9	43.0	67.6	49.3
Free Sugars *	PAHO	2.8 (1.8, 7.0)	11.2 (0.0, 2.3)	86.8 (0.4, 0.8)	14.8 (0.6, 1.4)	12 (1.0, 2.2)	0 (-)	57.8 (0.3, 0.6)	25.0 (5.6, 9.0)
	Chilean	22.5 (1.8, 2.7)	16.9 (0.8, 3.3)	84.4 (1.4, 2.6)	12.7 (1.5, 4.3)	14.2 (3.4, 7.0)	41.6 (5.0, 6.3)	66.2 (1.1, 2.0)	32.6 (2.1, 2.7)
	CATRP-FLNW	6.7 (1.8, 7.0)	11.2 (0.0, 4.6)	91.9 (0.5, 1.1)	24.6 (2.1, 3.6)	25.8 (2.7, 4.2)	14.5 (0.8, 1.2)	72.2 (0.9, 2.2)	34.2 (1.6, 2.3)
Mean		10.1	13.1	87.8	17.4	17.3	28.0	65.4	30.6
Saturated Fat	PAHO	94.2 (0.0, 0.0)	42.2 (0.4, 0.9)	33.3 (0.6, 1.1)	36.5 (0.5, 0.8)	89.5 (0.3, 0.4)	2.0 (-)	60.8 (0.2, 0.4)	60.5 (0.3, 0.4)
	Chilean	99.5 (0.0, 0.0)	98.5 (1.0, 1.4)	21.2 (0.7, 2.6)	28.1 (0.8, 1.6)	85.7 (0.7, 1.1)	47.9 (1.0, 2.0)	68.6 (0.6, 1.0)	65.1 (0.6, 0.8)
	CATRP-FLNW	98.5 (0.0, 0.0)	35.2 (0.0, 19.2)	32.3 (0.5, 1.0)	38.7 (6.0, 22.5)	89.5 (0.3, 0.4)	52.0 (0.4, 0.9)	60.8 (0.2, 0.4)	63.6 (0.3, 0.3)
Mean		97.4	58.6	28.9	34.4	88.2	34.0	63.4	63.1
Trans Fats	PAHO	94.7 (0.0, 0.0)	100 (-)	100 (0.0, 0.0)	98 (0.0, 5.7)	100 (0.0, 0.0)	18.7 (140.2, 207.8)	98.7 (0.0, 0.0)	94.9 (0.0, 0.0)
	Chilean	NA							
	CATRP-FLNW	99.1 (0.0, 0.0)	100 (-)	99.0 (0.0, 0.0)	93.2 (0.0, 0.1)	99.5 (0.0, 0.0)	95.8 (0.0, 11.8)	96.4 (0.0, 0.1)	97.2 (0.0, 0.0)
Mean		96.9	100	99.5	95.6	99.7	57.2	97.5	96.0

* Free Sugars for the PAHO-NPM, Total Sugars for Chilean-NPM and CATRP-FLNW. Percent of products that comply with the thresholds for critical nutrients, from the lowest to the highest confidence intervals (CI). NA: Not Applicable. Source: Elaborated by the authors.

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
