# Peer review of "Evaluation of the Nutritional Quality of Processed Foods in Honduras: Comparison of Three Nutrient Profiles"

_ijerph, 2020, doi:10.3390/ijerph17197060_

Round 1
Reviewer 1 Report
The objective of this study was to identify the proportion of processed foods, available in the Honduran marketplace, that exceed nutrient profile criteria relative to three recognized nutrient profile models. The authors focused on assessment of a set of critical nutrients, known to negatively influence human health by contributing to a diet that increases risk to chronic diseases and conditions. The purpose as I understand it, is to underscore the need for more explicit product labeling, to assist consumers to make healthier decisions, and the need for policies that could guide healthier food product development. The authors achieved the objective and purpose.
The authors generated a large amount of valuable, practical information and presented it in a variety of ways, that could be useful to address a variety of questions. Yet, there are several points throughout this manuscript that could be made clearer, easier for the reader:
- As a reader/reviewer, I would have liked to have had a better sense of the proportion of items in the complete dataset relative to the overall set of PF and UPF typically available in the supermarkets.
- Table 2: The logical flow of this table was not immediately apparent. It could be helpful to bold, shade, underline, or left-justify the first row of each category to indicate that it is the heading/umbrella term for that category. This would more readily convey that the remaining rows in that category are the types of items in that category. For example: Sweet snacks is the category heading; cookies, chocolates, candy are the types of products in that category. Here Sweet Snacks could be bolded, or the entire row could be shaded etc.
- Beginning on line 175, the authors discuss the geographical origins of the food products -- the highest and lowest rates. However, because there is only data for three locations, and given that the difference between the middle and lowest rates are minimal, a description of the highest and lowest is not very meaningful. Perhaps just a more general statement about the locations and the reference to Figure 2 would be more appropriate.
- Table 3 is on line 178, but the allusion to table 3 comes afterwards – line 183. It would be helpful to tie-in Table 3 to the flow of the manuscript prior to the table. (I understand this might already be the intent – but just in case, I wanted to put it on this list).
- Lines 182-184 describes Table 3 and suggests that “by the difference, [the reader could] estimate those found with excessive CN content.” It could be helpful to clarify by providing an example using one of the rows from the table. Lines 199 and 272 are good examples of integrating data or restating information for clarity.
- The two sentences spanning line 238 to line 241 are very confusing. I cannot tell with certainty, what is being said.
- The sentence that begins on line 230 should be restructured and streamlined for clarity.
Author Response
The proposal was reviewed for style and language by a professional editor.
Please see the attachment. Thank you for your comments.

Reviewer 2 Report
This study presents an interesting examination on the proportion of PF and UPF marketed in Honduras, which meet or not the criteria of three Nutrient Profile Models (NPM): PAHO (2016), Chile (2017) and the Central American Technical Regulation Proposal-Nutritional Warning Front Labeling (CATRP-NWFL 2017). However, the author should be to organize and summarize throughout manuscript.
The authors does not reported proper title, abstract and Introduction for what they intends to claim in the study. In fact, as the data of your results, there is no association between your hypothesis related results and obesity.
For easy viewing, please modify and add the footnotes in all of Table and Figure
Results section, please organize and summarize Fig or Tables of similar content into one paragraph.
Conclusions, Please modify conclusions based on your results. “The intervention of public health institutions and academia is essential to focus special attention on promoting FLNW: 1. Implement from the school stage basic models of nutrition and healthy lifestyles. 2. Use massive educational campaigns. 3. Support the research, strengthening, and updating of the Honduran food guide. 4. Training for artisanal producers in critical areas. As a recommendation for future studies, the sample could include sales, prices, and preferences in school booths and popular stores such as retail stores located in different neighborhoods and neighborhoods in the country.“
Author Response
The proposal was reviewed for style and language by a professional editor.
Please see the attachment. Thank you for your comments.

This manuscript is a resubmission of an earlier submission. The following is a list of the peer review reports and author responses from that submission.